# A Real-Time Semantic Segmentation Method Based on STDC-CT for Recognizing UAV Emergency Landing Zones

**DOI:** 10.3390/s23146514

**Published:** 2023-07-19

**Authors:** Bo Jiang, Zhonghui Chen, Jintao Tan, Ruokun Qu, Chenglong Li, Yandong Li

**Affiliations:** College of Air Traffic Management, Civil Aviation Flight University of China, Guanghan 618307, China; czh@cafuc.edu.cn (Z.C.); tjt@cafuc.edu.cn (J.T.); qurk@cafuc.edu.cn (R.Q.); lcl@cafuc.edu.cn (C.L.); yandongli@cafuc.edu.cn (Y.L.)

**Keywords:** real-time semantic segmentation, UAV, embedded device, emergency landing zones, deep learning

## Abstract

With the accelerated growth of the UAV industry, researchers are paying close attention to the flight safety of UAVs. When a UAV loses its GPS signal or encounters unusual conditions, it must perform an emergency landing. Therefore, real-time recognition of emergency landing zones on the ground is an important research topic. This paper employs a semantic segmentation approach for recognizing emergency landing zones. First, we created a dataset of UAV aerial images, denoted as UAV-City. A total of 600 UAV aerial images were densely annotated with 12 semantic categories. Given the complex backgrounds, diverse categories, and small UAV aerial image targets, we propose the STDC-CT real-time semantic segmentation network for UAV recognition of emergency landing zones. The STDC-CT network is composed of three branches: detail guidance, small object attention extractor, and multi-scale contextual information. The fusion of detailed and contextual information branches is guided by small object attention. We conducted extensive experiments on the UAV-City, Cityscapes, and UAVid datasets to demonstrate that the STDC-CT method is superior for attaining a balance between segmentation accuracy and inference speed. Our method improves the segmentation accuracy of small objects and achieves 76.5% mIoU on the Cityscapes test set at 122.6 FPS, 68.4% mIoU on the UAVid test set, and 67.3% mIoU on the UAV-City dataset at 196.8 FPS on an NVIDIA RTX 2080Ti GPU. Finally, we deployed the STDC-CT model on Jetson TX2 for testing in a real-world environment, attaining real-time semantic segmentation with an average inference speed of 58.32 ms per image.

## 1. Introduction

Due to the rapid advancement of unmanned aerial vehicle (UAV) technology, researchers across scientific and industrial disciplines have paid a great deal of attention to UAVs. Capturing and analyzing aerial images are important ways for UAVs to perceive their surroundings. This is significant in multiple fields, such as forest fire detection [1], vehicle detection [2,3], road construction [4], land cover [5,6,7], the oil and gas inspection industry [8,9], and traffic data analyses [10,11].

In UAV flight missions, recognizing emergency landing zones is crucial for ensuring UAV safety, especially when facing unexpected events such as GPS signal loss. To improve the accuracy of landing zone identification, researchers have recently adopted semantic segmentation methods. Semantic segmentation differs from traditional methods of UAV emergency landing zone identification, it assigns each pixel in the picture a label that makes sense, enabling precise identification of landing zones. This approach is particularly effective in densely populated urban scenes, where it can identify protected targets, such as people and vehicles, and reduce false alarms. Moreover, this method extracts semantic information of different objects in the landing zone, facilitating a better understanding of the structure and environment of emergency landing zones. Consequently, semantic segmentation has become an important tool to accurately identify landing zones.

The rapid evolution of deep learning has significantly improved the effectiveness of semantic segmentation. With the growth of numerous industrial domains, including autonomous driving [12,13], medical image diagnosis [14], and remote sensing [15], researchers no longer solely focus on accuracy improvement, they also pay more attention to the speed of segmentation. An efficient semantic segmentation model must have high segmentation accuracy, real-time capabilities, and a lightweight network framework in order to maintain the balance between accuracy and speed after the model has been deployed to an embedded device.

To meet these requirements, many researchers have started to study real-time semantic segmentation methods. Recently, some CNN-based methods have achieved low latency and high efficiency. For example, BiSeNet [16] proposes a dual-branch network topology that includes a small spatial pathway (SP) for preserving spatial information and generating high-resolution features, as well as a context pathway (CP) with a fast downsampling approach for obtaining an adequate receptive field. On these two pathways, a novel feature fusion module (FFM) is created to successfully recombine features. STDC-Seg [17] proposes a backbone network based on the short-term dense concatenate (STDC) module to extract deep features with scalable receptive fields and multi-scale information, building on the BiSeNet architecture. A Detail Guidance Module was also designed to encode spatial information in low-level features without an extra time-consuming approach, increasing the model’s inference speed.

Although the aforementioned models have demonstrated promising results in some circumstances, they may not perform well in real-world settings of unmanned aerial vehicle (UAV) aerial images because of high resolution, complex scenarios, and a large number of small objects [18]. Moreover, the limited computational resources of the onboard devices carried by UAV make it challenging to achieve real-time, high-quality semantic segmentation. Therefore, improving segmentation quality while achieving real-time performance has become a pressing issue in UAV semantic segmentation tasks.

In this paper, to identify suitable landing zones for UAVs, we mainly focus on improving the segmentation quality of various target objects in UAV remote sensing scenes while ensuring real-time segmentation. We designed the STDC-CT network based on the STDC1 backbone structure of STDC-Seg [17] due to the demonstrated efficiency of the STDC backbone network. Inspired by the multi-resolution attention mechanism [19], we propose the small object attention extractor (SOAE) module to focus on useful features in Stage 3–Stage 5 of the backbone network, especially for small object features. Attention weights of different levels are learned through a small network. Features from different layers are fused to enrich the information of small objects during feature extraction. We retain the Detail Guidance Module of STDC-Seg, but small object features are likely to be filtered out as noise during the model training. To solve the problem of insufficient noise resistance when extracting detail edges, we use the Laplacian of Gaussian (LoG) method [20,21,22] to replace the original Laplacian method. Additionally, we apply the parallel aggregation pyramid pooling module (PAPPM) [23] to increase the ability to gather multi-scale contextual information about the network and accelerate model inference speed. Finally, we design a detail and context feature fusion module (DCFFM) to fuse small object features, contextual information, and detail features. Finally, we use the small object attention feature (SOAE) branch to guide the feature fusion of the context and detail branches, allowing the model to effectively capture the detail information and global contextual information of small objects, improving the accuracy of recognizing small objects and STDC-CT performance.

This section provides a detailed introduction to the significance and research background of UAV emergency landing zones recognition. In Section 2, we present an overview of the related work in UAV emergency landing zones recognition, including the development history of traditional recognition methods, traditional semantic segmentation methods, and deep learning-based semantic segmentation methods. Section 3 provides a comprehensive description of the construction process of our proposed dataset, UAC-City. In Section 4, we explain the details of our proposed semantic segmentation method, STDC-CT. The effectiveness of our method is evaluated in Section 5 through extensive experiments. Results show that our proposed method has improved the performance of UAV emergency landing zones recognition compared with the state-of-the-art works.

The following are the key contributions of this paper:We create a new dataset for UAV aerial semantic segmentation and achieve recognition of emergency landing zones, protected targets, and buildings during high-altitude UAV flight missions.A lightweight semantic segmentation network named STDC-CT i is proposed for UAV emergency landing zones recognition. The proposed model consists of STDC backbone, SOAE module, PAPPM, DCFFM and Detail Guidance Module. The network achieves a balance of segmentation speed and accuracy and improves the segmentation accuracy on small objects.Extensive experiments have been carried out to evaluate the efficiency of our method. The results of our experiments indicate that our model can reach cutting-edge performance on the UAV-City, Cityscapes, and UAVid datasets. In addition, we deploy the trained model onto a UAV equipped with a Jetson TX2 embedded device. It shows that the model works well for real-world UAV applications.

## 2. Related Work

### 2.1. Conventional Methods for Recognizing UAV Emergency Landing Zones

Since the invention of UAVs, there has been continuous innovation in the methods used to recognize emergency landing zones, ranging from artificial methods to intelligent methods. Some of these methods are as follows [24,25,26,27]:Artificial recognition: the UAV pilot observes the surrounding environment and selects a suitable landing area, such as a flat and open grassland, ground, rooftop, etc.Photo analysis: this method involves analyzing the photos of the landing zones captured by a UAV camera to judge the flatness, obstacles, terrain, and other factors, thus aiding in landing zone selection.Altitude measurement: altitude measuring equipment mounted on the UAV can be used to measure the height of the landing zone and judge whether it meets safety requirements.Radar scanning: using radar equipment, the UAV scans the landing zones and obtains information such as the terrain height and obstacles to gauge the suitability of the landing area.GPS positioning: the GPS device mounted on the UAV obtains the location information of the landing zones, and evaluates factors such as terrain, height, and slope to comprehensively select a landing zone.Object recognition method: a deep learning-based pre-trained object detection model can be built on the UAV to process images or videos captured during flights in real time, thereby recognizing the positions of the landing zones.

While the above methods can meet the demands for recognizing drone landing zones under certain circumstances, they have certain limitations for special cases, such as loss of GPS signals or high-precision requirements for landing area recognition. To achieve more accurate recognition results, researchers have started to use semantic segmentation methods to recognize emergency landing zones for UAVs.

### 2.2. Conventional Semantic Segmentation Methods

Traditional semantic segmentation techniques were developed in the late 1980s and early 1990s, principally due to breakthroughs in image processing and computer vision technology. The development of traditional semantic segmentation algorithms can be summarized as follows:Threshold-based segmentation [28]: The earliest semantic segmentation algorithms were based on threshold segmentation, which involves categorizing pixels in an image into the foreground and background based on a fixed threshold and processing the foreground region further. However, this kind of method is not suitable for images with complex backgrounds.Region-based segmentation [29]: Region-based segmentation methods locate clusters of similar pixels in an image, which are considered to be of the same class of pixels. This method can handle images with complex backgrounds, but the results often contain discontinuous regions.Edge-based segmentation [30]: Edge-based segmentation methods categorize pixels into different classes by detecting edges in the image. This method can generate continuous segmentation results, but it is sensitive to noise.Graph-based segmentation [31,32]: Graph-based segmentation methods consider an image as a graph, where pixels are nodes and the edges represent the similarity between pixels. Segmentation is completed by optimizing the objective function. This method can produce highly accurate segmentation results, but it requires longer computation time and resources.Cluster-based segmentation [33]: Cluster-based segmentation methods categorize pixels in an image into different groups based on the similarity between pixels obtained through clustering. This method can handle large-scale images, but it is sensitive to noise.

Over the past few years, deep learning-based semantic segmentation methods have made significant progress, becoming mainstream semantic segmentation algorithms.

### 2.3. Methods for Semantic Segmentation Based on Deep Learning

#### 2.3.1. Generic Semantic Segmentation

Since the introduction of convolutional neural networks, methods based on fully convolutional networks (FCNs) have demonstrated outstanding performance in various benchmarks in the field of semantic segmentation. FCNs [34] were the first end-to-end, pixel-to-pixel trainable networks for semantic segmentation. FCNs also presented the first encoder–decoder structure in semantic segmentation, which is frequently used in subsequent networks. U-Net [35] improves both the precision and efficiency of semantic segmentation by using bilinear upsampling and skip connections based on FCNs. DeepLab [36] uses dilated convolutions to enlarge the receptive field and combines conditional random fields (CRFs) for post-processing to improve the segmentation accuracy. PSPNet [37] proposes a pyramid scene parsing network that utilizes pyramid pooling and cross-feature-map connections to achieve better contextual information utilization and segmentation results. RefineNet [38] achieves fine pixel-level segmentation through multi-level feature fusion and adaptive convolution. To further improve segmentation achievement, DANet [39] develops a dual attention mechanism network that makes use of both spatial and channel attention strategies. However, these methods often require extensive computational resources to achieve high segmentation accuracy due to high-resolution input images and complex network connections. These methods may not meet the requirements of semantic segmentation in real scenes, especially for devices with limited computing resources, such as UAVs.

#### 2.3.2. Lightweight Semantic Segmentation Network

Lightweight semantic segmentation networks typically use some lightweight backbone networks for feature extraction. Specifically, ResNet [40] is a traditional deep residual network that, by including residual blocks, addresses the issue of gradient disappearance and explosion in deep network training and facilitates network training. To minimize the number of parameters and compute additional complexity, some lightweight ResNet network variants have been proposed, such as MobileNetV2 [41] and ShuffleNetV2 [42], which use channel shuffle or depth-wise separable convolution techniques to optimize segmentation performance while minimizing calculation and parameter count. In addition, STDC [17] designed a new short-term dense concatenate module, to obtain variant scalable receptive fields with a small number of parameters. Moreover, integrating STDC into the U-net architecture, forming the STDC network, greatly improves the performance of semantic segmentation networks.

#### 2.3.3. Real-Time Semantic Segmentation

To meet the real-time requirements of semantic segmentation, ENet [43] uses smaller and fewer feature maps in the early stages of the network, greatly reducing the network parameters and improving its running speed. Through two processing streams—one carrying contextual information with full resolution for precise segmentation boundaries and the other passing through a series of pooling operations to produce high-level feature maps for recognition—FRRN [44] proposes a ResNet-like network architecture, combining multiple scales of contextual information and pixel-level precision. ICNet [45] compresses the PSPNet model and designs a cascaded feature fusion unit (CFF) and a cascaded label to guide model training to achieve fast semantic segmentation. EDANet [46] uses an asymmetric structure, combined with dilated convolutions and dense connections, to achieve efficient semantic segmentation with low computational costs and a small model size. BiSeNet [16] provides spatial pathway (SP) and contextual pathway (CP) to address the issues of spatial information loss and diminishing receptive fields, as well as the feature fusion module (FFM) and attention refinement module (ARM) to increase accuracy at a reasonable cost. ESNet [47] designed a set of factorized convolutional units (FCU) and their parallel counterparts (PFCU). In the parallel version, the residual module is designed using the transform–split–transform–merge technique, with the split branch employing dilated convolutions with varying ratios to widen the receptive field. In addition, the STDC-Seg [17] network removes the Spatial Path branch from BiSeNet, creates the Detail Guidance Module to guide model training, and designs an STDC backbone network to reduce the inference time of the model.

As shown in Table 1, we list the key innovations of the above methods and their specific performances on the Cityscapes dataset. It is not difficult to find that the above methods all aim to obtain high accuracy and high inference speeds at the cost of a small number of parameters. However, we find that it is not necessarily the case that the smaller the number of parameters in the model, the higher the inference speed, which is also inextricably linked to the network structure of the model. Therefore, in real-time semantic segmentation research, we can attempt to design novel lightweight network structures, such as multi-branch networks, to satisfy the balance between accuracy and inference speed. In semantic segmentation tasks, the trade-off between segmentation accuracy and inference speed refers to the maximization of the inference speed while maintaining acceptable segmentation accuracy. This allows the model to reflect good performance in embedded devices.

## 3. UAV-City Dataset

In our study, we collected aerial images captured by UAVs in Hangzhou, China, to construct our dataset. In the design of the entire process, we considered the practicality and effectiveness of research on the semantic segmentation of UAV aerial images. A total of 600 images with a resolution of 1280×720 were densely annotated for semantic segmentation tasks.

To enable the UAV to autonomously make timely landing decisions when receiving emergency landing instructions or encountering urgent situations, the UAV needs to recognize potential landing zones on the ground, as well as forced landing zones, protected targets, buildings, and other objects during high-altitude flights. Potential landing zones are defined as horizontal rooftops, horizontal grounds, and horizontal grasslands, while forced landing zones are defined as forests and rivers (lakes). Protected targets are defined as humans and vehicles. If no suitable zones nearby are available, then forced landing zones can be recognized for landing. During the landing process, UAVs need to continuously recognize pedestrians and vehicles on the ground to ensure the safety of lives and properties to the greatest extent possible.

### 3.1. Image Acquisition Strategy

During the operation of UAVs, strict compliance with safety regulations for drone flights is ensured.The maximum flight altitude of a drone is set at 140 m, with lateral flight maintaining stability at around 120 m.During image acquisition, the onboard camera captures continuous images of the ground with a time interval of 0.1 s, providing a top-down view. The camera angle is set vertically.Multiple flights are conducted to capture images from different flight paths, introducing variance into the dataset to mitigate the risk of overfitting during model training.Data collection is conducted under favorable weather conditions with sufficient daylight.

### 3.2. Image Processing and Annotation

#### 3.2.1. Image Filtering

As shown in Figure 1, the collected images consist of consecutive frames. To prevent overfitting and poor generalization during subsequent model training, multiple flights are conducted, and the images collected from each flight path are carefully selected. The selected images are annotated for semantic segmentation tasks, with a total of 600 images being annotated.

#### 3.2.2. Image Annotation

Our dataset is specifically designed for semantic segmentation tasks, but fully annotating all objects in urban aerial images is highly challenging. To achieve the recognition of potential landing zones, forced landing zones, protected targets, buildings, and other objects on the ground, we annotate the dataset with 11 categories, namely: horizontal roof, horizontal ground, horizontal lawn, river, plant, tree, car, human, building, road, and obstacle. The definitions of each category are as follows:Horizontal roof: the rooftop area of the buildings is flat.Horizontal ground: flat ground areas other than roadways used for vehicular traffic.Horizontal lawn: flat lawns.River: identifiable water bodies, including rivers and lakes.Plant: low vegetation, such as grass and shrubs.Tree: tall trees with canopies and trunks.Car: all vehicles on roads and parking lots, including cars, buses, trucks, tractors, etc.Human: all visible pedestrians on the ground.Building: residential buildings, garages, security booths, office buildings, and other structures under construction.Road: roads and bridges where vehicles are legally allowed to travel.Obstacle: steel frames, transmission line poles, and roads under construction.

The selected images are annotated with pixel-wise labeling by using the LabelMe annotation tool. Figure 2 shows an example of the annotation results. The labeled images of the original image were obtained by the generated JSON file, as shown in Figure 3.

### 3.3. Statistical Analysis

Our dataset contains 600 images of different scenes, which were annotated pixel by pixel using the LabelMe tool. Each image has a resolution of 1280×720. The dataset is divided into training, validation, and test sets according to an 8:1:1 ratio. Figure 4 shows the pixel counts of each class (including the background) in the UAV-City dataset. It clearly shows the distribution of imbalanced pixel counts among different classes. Most of the pixels come from the tree, road, and building classes. Classes such as car, horizontal roof, horizontal ground and horizontal lawn, river, obstacle, and plant contribute fewer pixels, accounting for less than 5% of the total pixels. For the human class, it occupies only 0.103% of the pixels, which is due to the relatively low number of pixels per instance in the UAV-City dataset. Among them, the car and human classes are small objects, which have fewer total pixel points and smaller sizes. Therefore, segmenting small objects poses a great challenge in semantic segmentation tasks.

## 4. Proposed Method

STDC-Seg [17] proposes a new STDC backbone network based on the BiSeNet [16] model, as shown in Figure 5a,b, which illustrates the layout of the STDC module, and Figure 5c presents the general STDC network architecture. Additionally, a new Detail Guidance Module is designed to replace the original Spatial Path branch of BiSeNet, which retains low-level detail features while reducing network computational complexity, resulting in an excellent real-time performance for semantic segmentation. However, this network still has some shortcomings in small object segmentation. Specifically, due to the FFM module gradually reducing the dimensionality of the feature maps, it may lose some detail information that is useful in small object segmentation tasks. Furthermore, the Detail Guidance Module in the STDC-Seg network mainly focuses on low-level feature layers, which may result in some detail information in high-level feature layers being ignored, affecting the accuracy of small object segmentation.

Therefore, in this section, we propose the STDC-CT network, as shown in Figure 6, which aims to improve small object segmentation accuracy while maintaining excellent segmentation speed.

### 4.1. Small Object Attention Extractor for STDC

We designed a small object attention extractor (SOAE) module for the STDC backbone network to extract more valuable features for small object recognition. We utilized convolutional layers at different resolutions (from Stage 3 to Stage 5) to extract diverse features from the input data. Each convolutional layer operates at a different resolution to capture features at different scales. Then, we used an attention mechanism to automatically select the most informative features and combine them for a more comprehensive representation.

In particular, we set the feature maps $F_{i}\left(i=1,2,3\right)$ generated by Stages 3 to 5 as the templates, denoted as $T^{t}\left(t=1,2,3\right)$, as illustrated in Figure 7. Then, the template layer $T^{t}$ was passed through a 1×1 convolutional layer with the dimension reduced to 1. The maximum value of the feature map was then generated through global max pooling, i.e., the most noteworthy feature pixels were captured; they were mapped to vectors for cosine similarity computation through a fully connected layer:(1)$$\begin{array}{c} {S_{t}}^{i}=Similarity\left(T^{t},T^{i}\right)=\frac{T^{t}\times T^{i}}{∥T^{t}∥\times ∥T^{i}∥}=\frac{\sum_{j=1}^{n}{T^{t}}_{j}\times {T^{i}}_{j}}{\sqrt{\sum_{j=1}^{n}{T^{t}}_{j}}\times \sqrt{\sum_{j=1}^{n}{T^{i}}_{j}}} \end{array}$$
where $S_{t}^{i}$ denotes the cosine similarity between the template layer $T^{i}$ and $T^{t}$. $T_{j}^{t}$ and $T_{j}^{i}$ denote the components of the vectors corresponding to $T^{t}$ and $T^{i}$, respectively. For example, when $T^{1}$ serves as the template, the cosine similarity values $S_{1}^{2}$ between $T^{2}$ and $T^{1}$, and $S_{1}^{3}$ between $T^{3}$ and $T^{1}$, indicate the attention levels of $T^{2}$ and $T^{3}$ toward $T^{1}$, respectively, while $S_{1}^{1}=1$. Similarly, when $T^{2}$ serves as the template, the cosine similarity values $S_{2}^{1}$ between $T^{1}$ and $T^{2}$, and $S_{2}^{3}$ between $T^{3}$ and $T^{2}$, indicate the attention levels of $T^{1}$ and $T^{3}$ toward $T^{2}$, respectively, while $S_{2}^{2}=1$. To evaluate the attention weights $a_{t}^{i}$, we input $\left\{S_{t}^{1},S_{t}^{2},S_{t}^{3}\right\}$ into the softmax layer to obtain normalized weight values:(2)$$\begin{array}{c} {a_{t}}^{i}=\frac{\exp({S_{t}}^{i})}{\sum_{i=1}^{3}\exp({S_{t}}^{i})} \end{array}$$

After obtaining the attention weights, the feature map sizes $F_{2}$ and $F_{3}$ must be constant with that of $F_{1}$; thus, we unsampled $F_{2}$ and $F_{3}$ by 2× and 4×, respectively, and the number of channels of $F_{1}$, $F_{2}$, and $F_{3}$ were 256, 512 and 1024 respectively. Then we added the weighted feature maps of the three levels, element by element, and adjusted the number of channels by a 1 × 1 convolution to obtain the attention feature:(3)$$\begin{array}{c} A_{t}=\sum_{i=1}^{3}{a_{t}}^{i}\times F_{i} \end{array}$$
where $A_{t}$ denotes the attention feature obtained through the network with $T^{t}$ as the template layer.

### 4.2. Laplacian of Gaussian for Detail Guidance Module

The Laplacian convolution method was employed in the STDC-Seg network for extracting fine details of image edges. The Laplacian operator, being an excellent edge detection operator, has been widely utilized in edge detection tasks. The Laplacian operator is commonly used in edge detection to detect significant changes in pixel intensity. However, with aerial images captured by drones, which often contain numerous small objects, such as vehicles and pedestrians, the edges of these small objects typically exhibit smaller intensity variations. As a result, when applying the Laplacian operator, the edges of small objects may be erroneously enhanced as noise. Additionally, since noise can be present in various locations within an image, the Laplacian operator tends to respond strongly to noise. When the operator is applied to pixels surrounding small objects, the noise can cause a strong response, leading to the edges of small objects being overshadowed by the noise signal. Consequently, in edge detection of aerial images captured by drones, the edges of small objects may be mistakenly considered as noise and consequently removed.

As shown in Figure 8, to address these issues and preserve the excellent edge extraction capability of the Laplacian operator, we apply the Laplacian of Gaussian (LoG) convolution method [21] in the Detail Guidance Module. Unlike the Laplacian convolution method, the LoG convolution method first applies Gaussian filtering to the image, which blurs the high-frequency details and reduces the intensity of noise. By separating the edges of small objects from the noise, the influence of noise is reduced. Additionally, Gaussian filtering reduces the gradient variations along the image edges. The variations of edge gradients are highly responsive to the Laplacian operator; noise typically has a high-frequency component and responds strongly to the Laplace operator. Therefore, by reducing the variations in edge gradients, Gaussian filtering can decrease the interference of noise in edge detection, enabling easier detection of the edges of small objects. Images are processed by a Gaussian filter and the Laplacian convolution, which preserves the outstanding edge extraction capability of the Detail Guidance Module while suppressing the impact of noise. This method can improve the accuracy of edge detail extraction. To reduce computational overhead, we leverage the associativity property of convolution operation, where the Gaussian function is combined with the Laplacian operator to form a single convolution kernel. As a result, only one convolution operation is needed on the image. The process of combining Gaussian and Laplacian methods is outlined below:

The Gaussian equation is:(4)$$\begin{array}{c} gauss\left(x,y,\sigma \right)=\frac{1}{2\pi \sigma^{2}}\exp\left(-\frac{x^{2}+y^{2}}{2\sigma^{2}}\right) \end{array}$$

Performing the Laplacian transformation on the two-dimensional Gaussian function:(5)$$\begin{array}{c} \begin{array}{cc} \nabla^{2}\left(gauss\left(x,y,\sigma \right)\right) & =\frac{\nabla^{2}\left(gauss\left(x,y,\sigma \right)\right)}{\partial^{2}x}+\frac{\nabla^{2}\left(gauss\left(x,y,\sigma \right)\right)}{\partial^{2}y}\\ \; & =\frac{1}{2\pi \sigma^{2}}\frac{\partial (-\frac{x}{\sigma^{2}}\exp\left(-\frac{x^{2}+y^{2}}{2\sigma^{2}}\right))}{\partial x}+\frac{1}{2\pi \sigma^{2}}\frac{\partial (-\frac{y}{\sigma^{2}}\exp\left(-\frac{x^{2}+y^{2}}{2\sigma^{2}}\right))}{\partial y}\\ \; & =\frac{1}{2\pi \sigma^{4}}\left(\frac{x^{2}}{\sigma^{2}}-1\right)\exp\left(-\frac{x^{2}+y^{2}}{2\sigma^{2}}\right)+\frac{1}{2\pi \sigma^{4}}\left(\frac{y^{2}}{\sigma^{2}}-1\right)\exp\left(-\frac{x^{2}+y^{2}}{2\sigma^{2}}\right)\\ \; & =\frac{1}{2\pi \sigma^{4}}\left(\frac{x^{2}+y^{2}}{\sigma^{2}}-2\right)\exp\left(-\frac{x^{2}+y^{2}}{2\sigma^{2}}\right) \end{array} \end{array}$$
where *x* and *y* denote the image pixel coordinates, $\nabla^{2}\left(gauss\left(x,y,\sigma \right)\right)$ denotes the LoG operator.

### 4.3. PAPPM for Capturing Contextual Information

To enhance the neural network’s ability to capture multi-scale contextual information while maintaining high efficiency, researchers proposed a technique named parallel aggregation pyramid pooling module (PAPPM) [23]. The PAPPM module consists of four parallel pyramid pooling branches, each using a different pooling kernel size to extract pooling features from feature maps at different scales. These features are then concatenated together to effectively capture multi-scale contextual information. Meanwhile, to accelerate the inference speed, the module adopts parallel computation, reducing the computational overhead while ensuring accuracy. The PAPPM module is integrated into the STDC network, as shown in Figure 9.

### 4.4. The Detail and Context Feature Fusion Module

In the STDC-CT network, the context branch contains rich semantic information that can provide more accurate semantic representations. However, it loses a significant amount of spatial and geometric information due to the continuous downsampling process. On the other hand, in order to balance the detail information and contextual information in the STDC-CT network, we utilize the small object attention features extracted by the previously designed SOAE module to guide the fusion of edge and detail features with contextual information. This is because one of the original intentions of designing the STDC-CT network is to improve the accuracy of small object segmentation in drone imagery. Therefore, we have high confidence in the small object attention branch and use it as a guiding variable to balance the fusion of the detail branch and contextual information. Specifically, in the proposed approach, the attention feature is first passed through a sigmoid layer for normalization to obtain the weight value ω, where ω denotes the weight parameter for the detail features and 1−ω denotes the weight parameter for the contextual information. Next, the detail features and contextual information are multiplied with their corresponding weight values and then added together to perform feature fusion. Finally, the features are processed through a CONV-BN-ReLU to obtain $F_{out}$. The detailed structure of the method is illustrated in Figure 10.

The small object attention branch, the context branch, and the detail aggregation module branch, respectively, are denoted as *a*, *c*, and *d*. These branch results are shown as vectors of corresponding pixels with the symbols ${\vec{p}}_{a}$, ${\vec{p}}_{c}$, and ${\vec{p}}_{d}$, respectively. The representations of ω and $F_{out}$ can be written as follows:(6)$$\begin{array}{c} \omega =Sigmoid({\vec{p}}_{a}) \end{array}$$
(7)$$\begin{array}{c} F_{out}=ConvX\left(\left(1-\omega \right)⊗{\vec{p}}_{c}+\omega ⊗{\vec{p}}_{d}\right) \end{array}$$
where ConvX consists of a layer for convolution, a layer for batch normalization, and a layer for ReLU activation. When ω>0.5, the model’s training places higher confidence in the detail features, whereas when ω<0.5, the model’s training places higher confidence in the contextual information.

## 5. Experimental Results

We undertake model training on the UAV-City, Cityscapes [48], and UAVid [18] dataset to verify the efficacy of our proposed STDC-CT approach. We compare our model with other mainstream semantic segmentation models and advanced real-time networks. Finally, we deploy the STDC-CT model on a TX2-equipped UAV for real-world environment testing.

### 5.1. Datasets

**UAV-City.** The details of the UAV-City dataset can be found in Section 3 of this paper. The dataset consists of 600 finely annotated aerial images captured by UAV, with each image having a resolution of 1280×720. A total of 11 categories are annotated for the task of semantic segmentation.

**Cityscapes.** Cityscapes [48] is a widely used public dataset for computer vision research, containing a large collection of urban street scene images and corresponding annotated data. The Computer Vision study group at the University of Stuttgart, Germany, created this dataset with the goal of providing high-quality data resources for studying in fields such as autonomous driving, traffic planning, and urban development. The dataset includes 5000 high-resolution images covering various street scenes in 50 cities, which are divided into training, validation, and testing sets with 2975, 500, and 1525 images, respectively. The annotations include 30 categories, but only 19 categories are used for semantic segmentation. The image resolution is 2048×1024, which requires real-time segmentation performance.

**UAVid.** UAVid [18] is a widely used dataset for semantic segmentation in the context of UAV. The dataset includes 30 video sequences captured from a tilted viewpoint, with 4K high-resolution images. A total of 300 images are densely annotated for 8 categories, including building, road, static car, tree, low vegetation, humans, moving car, and background clutter, for the task of semantic segmentation. The image resolutions are either 4096×2160 or 3840×2160.

### 5.2. Implementation Details

**Training.** With a weight decay of $5\times 10^{-4}$ and a momentum of 0.9, the stochastic gradient descent (SGD) algorithm is employed as the optimizer. Considering the differences in datasets, we devised different training strategies for each dataset. The batch size for Cityscapes is 24, the maximum number of iterations is 120,000, and the initial learning rate is 0.01. The batch size for UAVid is set to 8, the maximum number of iterations to 10,000, and the initial learning rate at 0.004. The batch size for UAV-City is set to 16, the maximum number of iterations to 10,000, and the initial learning rate is 0.001. We configured the experimental environment with PyTorch-1.12.1 in Anaconda and executed all experiments on an NVIDIA RTX 2080Ti GPU with CUDA 11.6 and CUDNN 8.5.0. We utilized the “poly” learning rate methodology, where the learning rate varies based on the following equation:(8)$$\begin{array}{c} lr_{t}=lr_{0}\times {\left(1-\frac{iter_{c}}{iter_{\max}}\right)}^{p} \end{array}$$
where $lr_{t}$ denotes the current learning rate, ${lr}_{0}$ denotes the initial learning rate, ${iter}_{c}$ denotes the current iteration count, ${iter}_{max}$ denotes the maximum iteration count, and *p* denotes the power, which is set to 0.9 in this case.

**Evaluation Metric.** To compare the test results of the model with the ground truth labels, we obtain a confusion matrix $P=\left\{p_{ij}\right\}\in N^{\left(m+1)\times (m+1\right)}$, where $p_{ij}$ denotes the number of pixels belonging to class *i* but classified as class *j*. The total number of classes, including *k* classes and one background class, is denoted as m+1. Specifically, the diagonal elements $p_{ij}$ denote the number of pixels with correct predictions. To evaluate the effectiveness of the segmentation results, we mainly use mIoU as the evaluation metric. This is because mIoU is invariant to the sizes of different objects and provides more comprehensive information compared to other metrics, such as accuracy, precision, and recall; mIoU takes into account both the accuracy and precision of classification, making it more suitable for the UAV-City dataset proposed in this paper. The mIoU calculating formula is as follows:(9)$$\begin{array}{c} \frac{1}{m+1}\sum_{i=1}^{m+1}\frac{p_{ij}}{\sum_{j=1}^{m+1}p_{ij}+\sum_{j=1}^{m+1}p_{ji}-p_{ii}} \end{array}$$

### 5.3. Ablation Study

Our designed STDC-CT network consists of multiple modules with different functionalities. To verify their effectiveness, ablation experiments were conducted in this study using a 0.75 input scale, with STDC-Seg75 serving as the baseline network. The experimental results are shown in Table 2 and Table 3, where Table 2 denotes the mIoU values of different templates when using the SOAE module on the Cityscapes and UAV-City test sets based on the baseline network. The trial results show the utilization of $T^{3}$, which corresponds to the feature map $F_{3}$ output from Stage 5, as the template achieves the best performance. Table 3 denotes the experimental results of adding different modules on top of the baseline network. SOAE refers to the small object attention extractor, LoG refers to the method with the Laplacian of Gaussian, PAPPM refers to parallel aggregation PPM, and DCFFM refers to the detail and context feature fusion module. The ablation experimental results indicate that each module proposed in our STDC-CT network contributes to the improvement of semantic segmentation accuracy. Specifically, the SOAE module in STDC-CT improved mIoU by 0.4% and 0.7% on the CityScapes and UAV-City datasets, respectively. The LoG module increased mIoU by 0.2% and 0.4% on the same datasets, while the PAPPM module resulted in mIoU improvements of 0.3% and 0.4%. Moreover, the DCFFM module led to mIoU improvements of 0.3% and 0.7% on the CityScapes and UAV-City datasets, respectively. In summary, the proposed modules in STDC-CT are effective for semantic segmentation.

### 5.4. Compare with Mainstream Methods

On the UAV-City, Cityscapes, and UAVid datasets, we compare our approach with mainstream methodologies in this section.

**Results on UAV-City.** We compare our proposed STDC-CT network with U-Net [35], PSPNet [37], DeepLabv3+ [49], STDC-Seg [17], and our STDC-CT on the UAV-City test set. As shown in Table 4, our STDC-CT network achieves a mIoU of 67.3% with an input scale of 0.75, corresponding to an image input size of 960×540, and operates at 196.8 FPS. This performance surpasses STDC-Seg by 2.2%. Building upon these results, we further generate detailed IoU values for each class in the UAV-City dataset, as shown in Table 5. Our method achieves the highest IoU values in most classes, outperforming other methods. Additionally, we visualize the attention mechanisms of STDC-Seg and STDC-CT models for the car class by using Grad-CAM, as shown in Figure 11. Where (a) shows the input image, (b) shows the results from STDC-Seg, and (c) shows the results from STDC-CT. It can be observed that our STDC-CT model exhibits more focused attention on the car, reducing false positives. Therefore, our STDC-CT model demonstrates significant advantages in small object segmentation.

**Results on Cityscapes.** On the Cityscapes test set, we assign the segmentation accuracy and inference speed of the method proposed in Table 6. Compared to previous methods, our method improves segmentation accuracy with minimal loss in inference speed. At an input scale of 0.75, corresponding to an image size of 1536×768, our STDC-CT network achieves a mIoU of 76.5% at a speed of 122.6 FPS, outperforming STDC-Seg75 by 1.2%.

**Results on UAVid.** As shown in Table 7, we conducted experiments on the UAVid dataset using mainstream semantic segmentation methods as well as our proposed STDC-CT method. The IoU values of each class are measured on the test set. The experimental results indicate that STDC-CT achieves the highest IoU values in the clutter, building, tree, and moving car classes, and the mIoU value of STDC-CT is 2.5% higher than the STDC-Seg.

### 5.5. Embedded Experiments

The TX2 [59] is an embedded computing board based on NVIDIA’s Tegra X2 system-on-chip (SoC), known for its high performance, low power consumption, and powerful computing capabilities. It is commonly used in sectors such as robots, self-driving cars, and intelligent edge computing. The detailed specifications of the NVIDIA TX2 are shown in Table 8.

The NVIDIA TensorRT inference acceleration library was utilized to maximize GPU resource utilization on the TX2. We configured JetPack 4.5.1 and CUDA 10.2 on the NVIDIA TX2, and set up the experimental environment with PyTorch 1.9.1. Then, we deployed the trained models on the TX2 for real-time segmentation testing in the UAV-City dataset scenarios, with each image input size set to 960×540. Table 9 presents the inference speeds of various models for predicting a single image. The experimental results show that our proposed STDC-CT model achieves an average inference speed of only 58.32 ms on the TX2, effectively meeting the real-time segmentation demands in real-world environments.

### 5.6. Analysis of UAV Emergency Landing Zone Recognition Results

Our model produced high-quality segmentation results on the UAV-City test set. Figure 12 shows the visual results of our experiments, which demonstrate that our STDC-CT method can accurately identify the emergency landing zones for UAVs as well as other objects on the ground, such as the car and human classes. In comparison, the segmentation results of other methods do not exhibit the same level of superior performance in terms of fine texture details and accurate small object recognition as STDC-CT.

Finally, we mounted the TX2 onto the UAV for real-world flight testing. As shown in Figure 13a–c, we present the results of recognizing the horizontal roof, ground, and lawn using the STDC-CT method, respectively. The experimental results show that our method can successfully recognize UAV emergency landing zones in real environments, thus fully validating the superiority and practicality of our method.

## 6. Discussion

Recognizing emergency landing zones is extremely important for unmanned aerial vehicles (UAVs) as they may encounter unexpected situations during flight missions. Compared with traditional methods, utilizing semantic segmentation techniques for recognizing emergency landing zones can provide higher accuracy; this can ensure UAV flight safety and protect the safety of ground vehicles and pedestrians. As described in Section 3, due to the complex scenes and confusing backgrounds of UAV aerial images, UAVs face great challenges when performing semantic segmentation tasks. Our UAV-City dataset provides a wider field of view for UAVs to choose landing zones. However, due to the relatively high altitudes at which drones operate, the visibility of small things, like people on the ground, is constrained. Therefore, during the actual landing processes, the real-time segmentation of UAV aerial images is necessary. As shown in Table 1, Figure 11 and Figure 12, the proposed STDC-CT model captures rich semantic information in global contextual information and obtains multi-scale features through different operations, which is important for small object segmentation. However, while the proposed network can achieve excellent segmentation results and provide additional technical support for UAVs in recognizing emergency landing zones, UAVs may still encounter more complex ground environments during real-time operations. Therefore, comprehensive landing strategies need to be further refined based on new factors. Ultimately, the unconditional protection of lives and property on the ground should be ensured.

## 7. Conclusions

In this paper, we established a new UAV aerial dataset for studying the recognition of UAV emergency landing zones. The dataset consists of 600 images with a capture height of 120 m and a size of 1280×720 pixels. Each image is densely annotated with 12 categories for semantic segmentation experiments. To adapt to the characteristics of UAV aerial images, we propose the STDC-CT model, which is based on the STDC backbone network for the real-time semantic segmentation of UAV scenes. The advantage of this method is that it can improve the segmentation accuracy of small objects without generating a large amount of redundant computations. Extensive experiments and visualization results demonstrate the effectiveness of the STDC-CT network. On the UAV-City dataset, we achieve 67.3% mIoU at a speed of 196.8 FPS, 76.5% mIoU at a speed of 122.6 FPS on the CityScapes dataset, and 68.4% mIoU on the UAVid dataset, while also improving the segmentation accuracies of vehicles and pedestrians in UAV scenes. Moreover, the inference speed for each image on the NVIDIA TX2 embedded device is only 58.32 ms. Finally, we mounted the TX2 onto the UAV for real-world flight testing and successfully recognized emergency landing zones. In the future, we will continue to explore model lightweight model techniques to further improve segmentation performance.

## Figures and Tables

**Figure 1 sensors-23-06514-f001:**
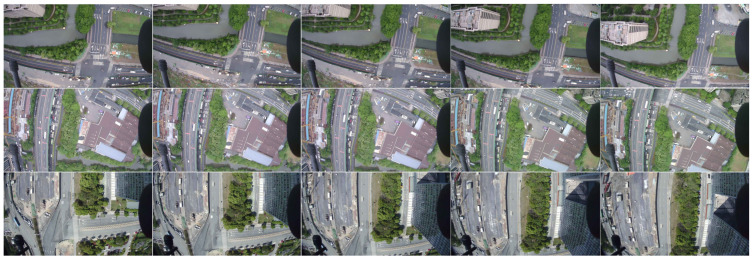
The aerial images are captured in consecutive frames.

**Figure 2 sensors-23-06514-f002:**
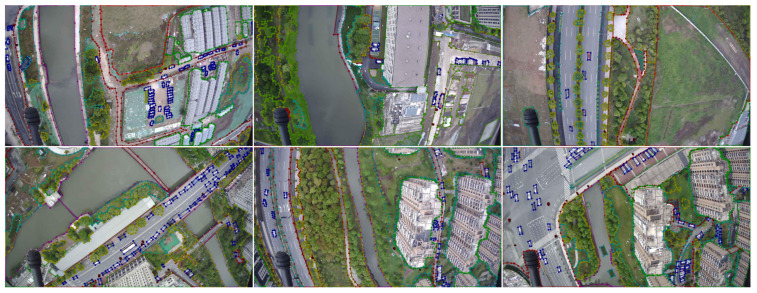
Example of the image annotation.

**Figure 3 sensors-23-06514-f003:**
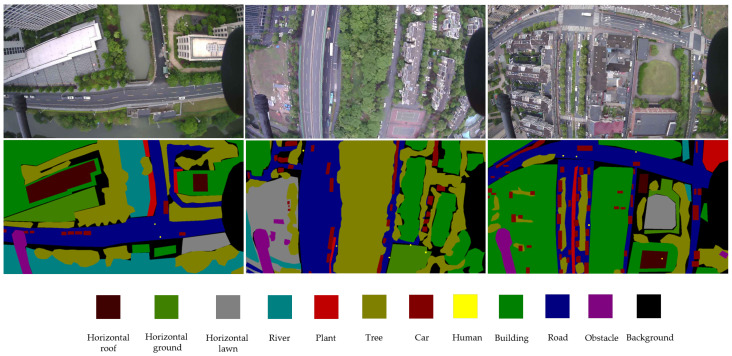
Images and labels from the UAV-City dataset are provided as examples. The photographs obtained by the UAV are shown in the first row. The ground truth labels are shown in the second row.

**Figure 4 sensors-23-06514-f004:**
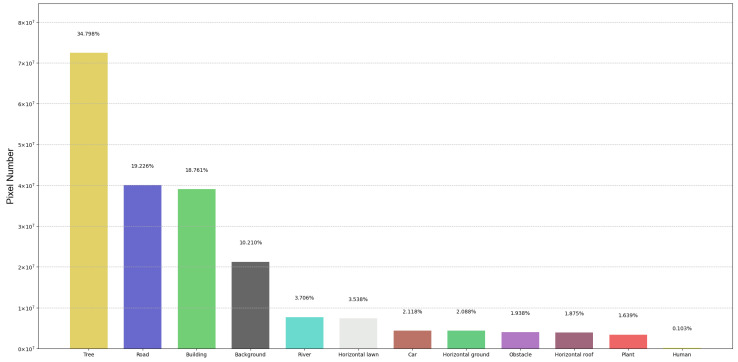
Pixel number and histogram of UAV-City.

**Figure 5 sensors-23-06514-f005:**
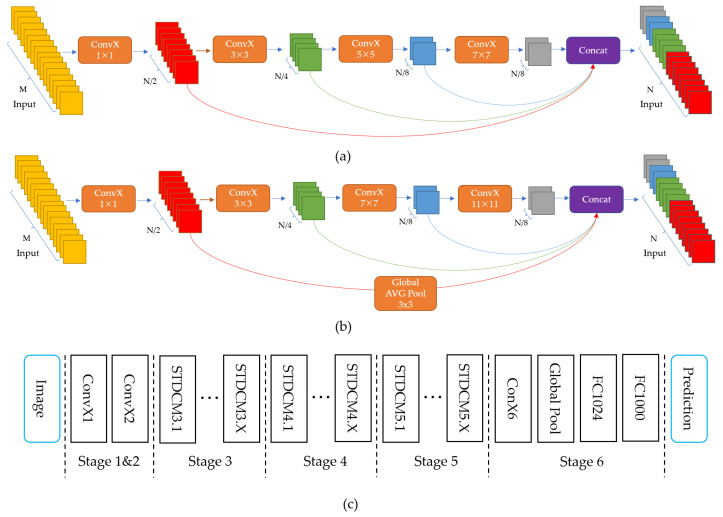
The STDC network. (**a**) Stride = 1; (**b**) stride = 2; (**c**) shows the General STDC Network architecture. The ConvX operation denotes Conv-BN-ReLU.

**Figure 6 sensors-23-06514-f006:**
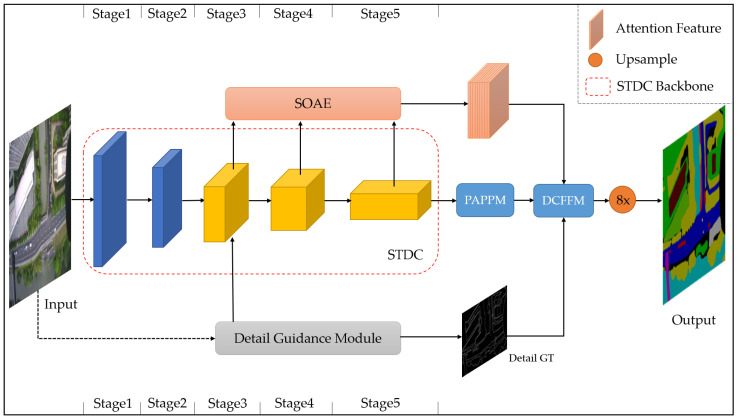
The STDC-CT segmentation network we propose.

**Figure 7 sensors-23-06514-f007:**
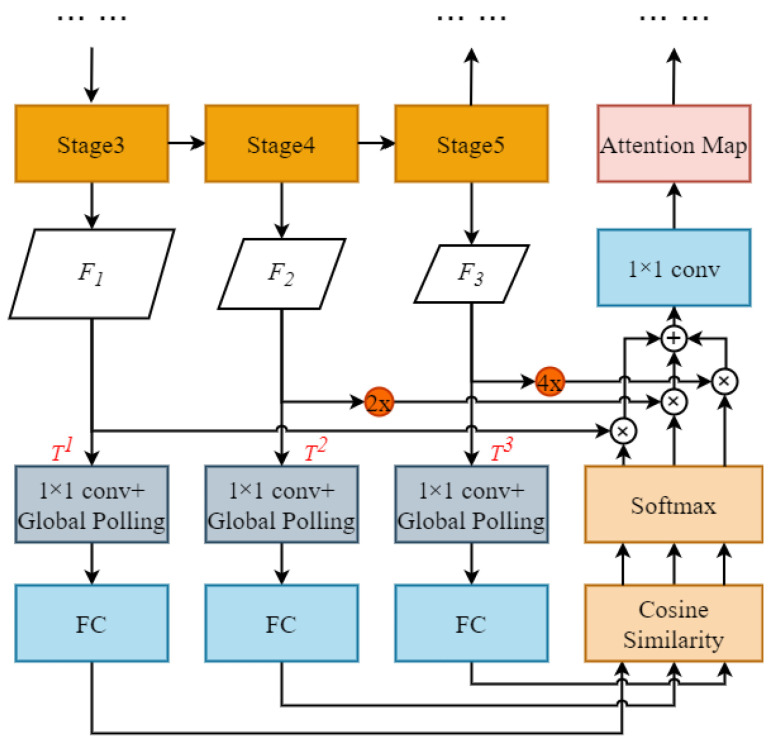
The small object attention extractor module.

**Figure 8 sensors-23-06514-f008:**
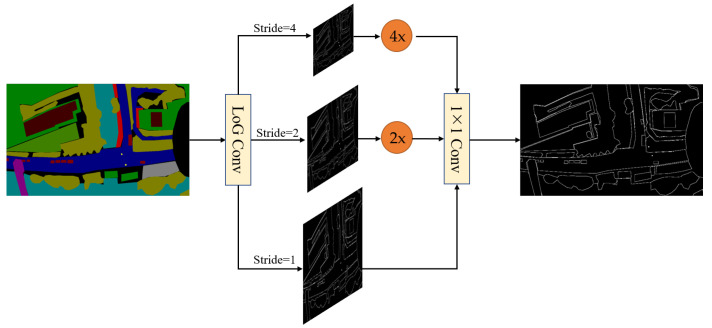
The Detail Guidance Module. The LoG ConV denotes the Laplace of Gaussian convolution, which generates soft thin detail feature maps by convolution operations with different strides, thus obtaining multi-scale detail information. These detail features are then upsampled and mapped to the same sizes as the detail features after 1 × 1 convolution, and are then dynamically weighted to obtain the ground truth with binary detail.

**Figure 9 sensors-23-06514-f009:**
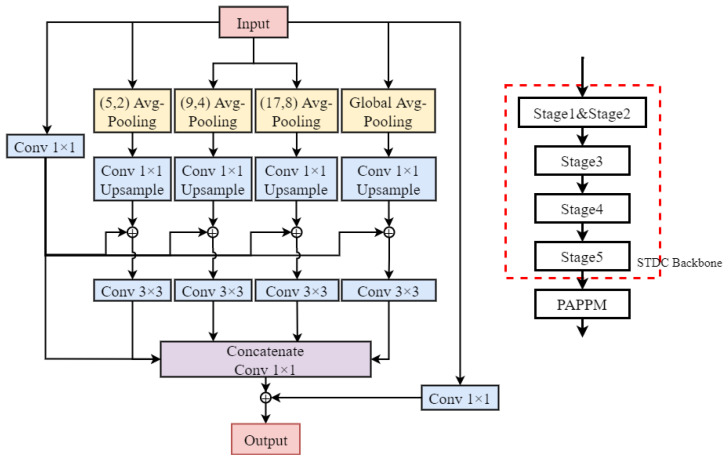
The PAPPM network architecture.

**Figure 10 sensors-23-06514-f010:**
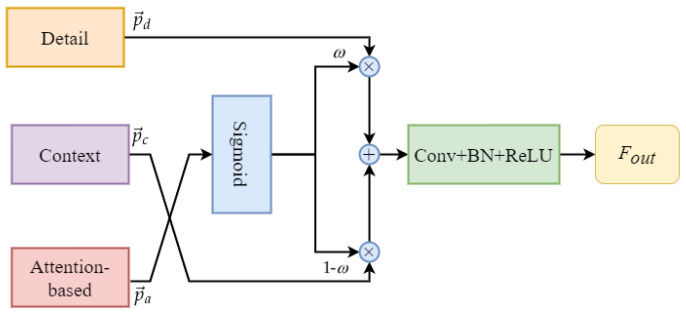
Detail and context feature fusion module.

**Figure 11 sensors-23-06514-f011:**
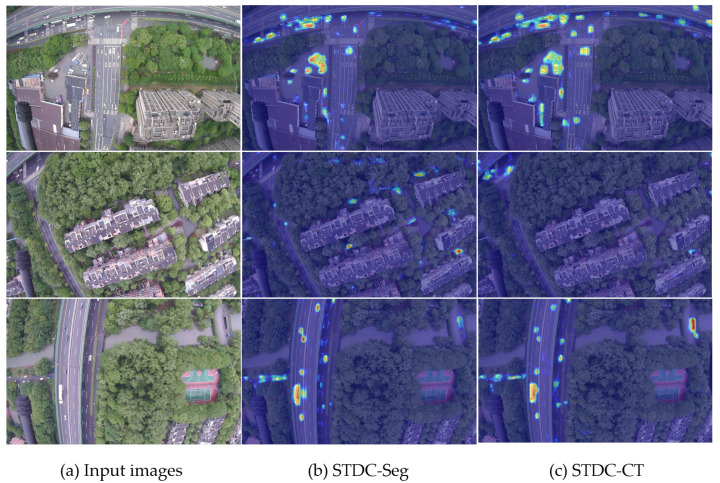
Comparison of Grad-CAM between STDC-CT and STDC-Seg on the UAV-CITY dataset.

**Figure 12 sensors-23-06514-f012:**
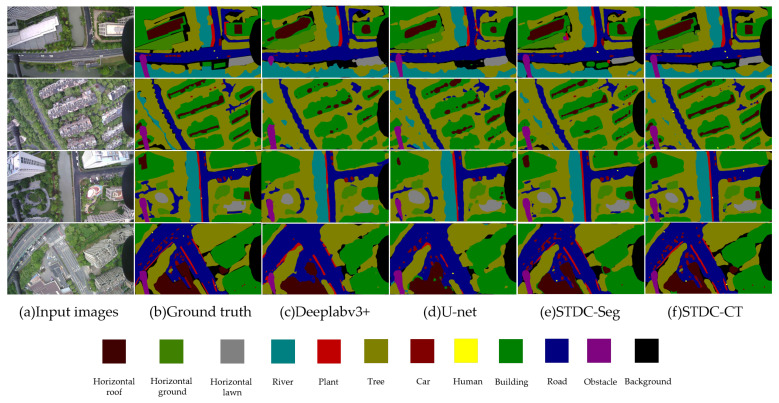
Semantic segmentation results on the UAV-City dataset with DeepLabv3+, U-net, STDC-Seg, and STDC-CT, respectively.

**Figure 13 sensors-23-06514-f013:**
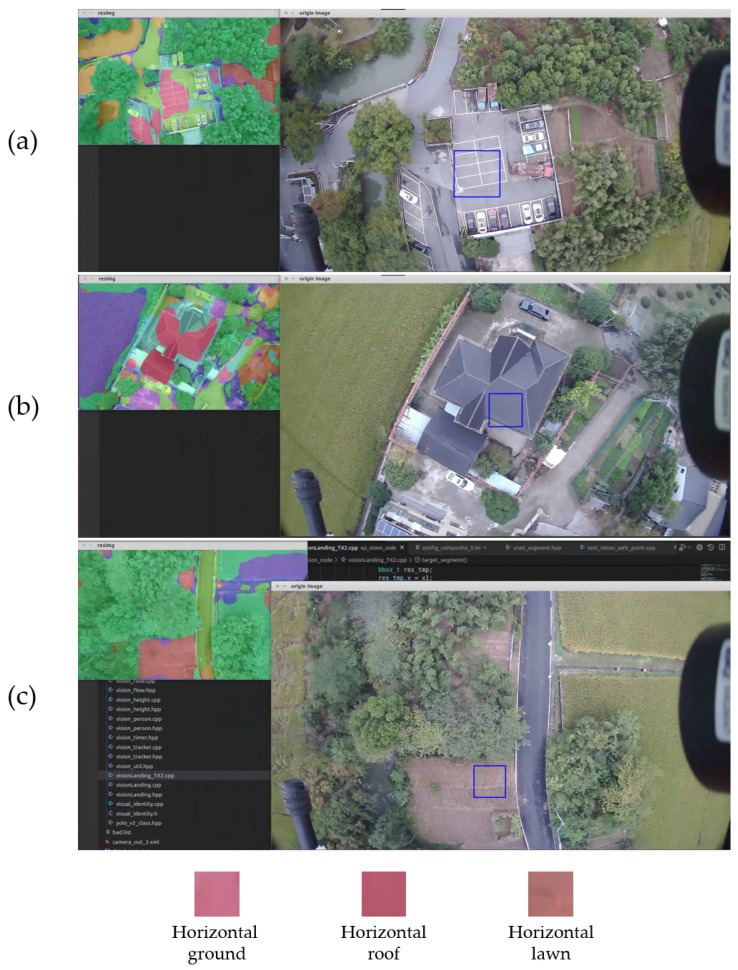
Results of real-world flight testing. We have recognized the horizontal ground as the emergency landing zone in subfigure (**a**). Subfigures (**b**,**c**) show that we have recognized the horizontal roof as the emergency landing zone and the horizontal lawn as the emergency landing zone, respectively.

**Table 1 sensors-23-06514-t001:** The state-of-the-art methods related to real-time semantic segmentation.

Model	Backbone	Key Innovations	mIoU (%)	FPS	Parameters
ENet [43]	-	Initial Block, Bottleneck Module	58.3	76.9	0.37 M
FRRN [44]	ResNet	FRRU, Full-resolution Residual Networks (FRRNs)	71.8	2.1	-
ICNet [45]	PSPNet-50	Cascade Feature Fusion Unit (CFF), The Loss Function	69.5	30.3	-
EDANet [46]	-	EDA Module	67.3	108.7	0.68 M
BiSeNet [16]	ResNet-18	Attention Refinement Module (ARM), Feature Fusion Module (FFM)	74.7	65.5	49.0 M
ESNet [47]	-	factorized convolutional units (FCU) and their parallel counterparts (PFCU)	70.7	63	1.66 M
STDC-Seg75 [17]	STDC1	Short-Term Dense Concatenate Module (STDC), Detail Guidance Module	75.3	126.7	-

**Table 2 sensors-23-06514-t002:** Performance of the SOAE module with different templates.

Template	mIoU (%)	
	Cityscapes	UAV-City
SOAE-$T^{1}$	74.8	64.7
SOAE-$T^{2}$	75.4	65.4
SOAE-$T^{3}$	**75.7**	**65.8**

**Table 3 sensors-23-06514-t003:** Study of ablation for our proposed modules on the Cityscapes and UAV-City test sets.

Method	SOAE	LoG	PAPPM	DCFFM	mIoU (%)	
					UAV-City	Cityscapes
STDC-Seg75 [17]	-	-	-	-	65.1	75.3
STDC-CT75	✓	-	-	-	65.8	75.7
STDC-CT75	✓	✓	-	-	66.2	75.9
STDC-CT75	✓	✓	✓	-	66.6	76.2
STDC-CT75	✓	✓	✓	✓	**67.3**	**76.5**

**Table 4 sensors-23-06514-t004:** Comparisons with other mainstream methods on UAV-City.

Model	Resolution	Backbone	mIoU (%)	FPS
U-Net [35]	960×540	VGG16	63.4	28.9
PSPNet [37]	960×540	ResNet50	52.1	34.5
DeepLabv3+ [49]	960×540	MobileNetV2	57.4	35.5
STDC-Seg [17]	960×540	STDC1	65.1	**212.3**
STDC-CT	960×540	STDC1	**67.3**	196.8

**Table 5 sensors-23-06514-t005:** The results of the experiment on the UAV-City dataset.

Model	Class IoU (%)												mIoU (%)
	Hor. Roof	Hor. Gro.	Hor. Lawn	River	Plant	Tree	Car	Hum.	Bui.	Road	Obs.	Back.	
U-Net	63.4	51.5	57.7	81.4	57.9	85.6	56.1	13.5	78.9	80.6	81.5	53.5	63.4
PSPNet	52.1	47.8	55.2	75.3	43.5	80.6	32.2	2.1	71.7	73.2	68.8	51.5	54.5
DeepLabv3+	54.1	60.2	65.2	77.8	36.2	84	42.4	5.8	76.2	75.2	65.8	50	57.7
STDC-Seg75	64.3	**62.8**	58.5	80.6	**60.5**	83.4	58.1	16.1	81.6	79.5	**83.9**	52.3	65.1
STDC-CT75	**65.6**	62.3	**66.2**	**85.7**	59.2	**86.7**	**61.3**	**18.6**	**83.1**	**82.5**	82.1	**54.3**	**67.3**

**Table 6 sensors-23-06514-t006:** Comparisons with other mainstream methods on Cityscapes.

Model	Backbone	GPU	mIoU	FPS
ENet [43]	-	Nvidia Titan X	58.3	76.9
ICNet [45]	PSPNet-50	Nvidia Titan X	69.5	30.3
HMSeg [50]	-	GTX 1080Ti	74.3	83.2
BiSeNetV1 [16]	ResNet-18	GTX 1080Ti	74.7	65.5
SwiftNet [51]	ResNet-18	GTX 1080Ti	75.4	39.9
HyperSeg-M [52]	EfficientNet	GTX 1080Ti	75.8	36.9
BiSeNetV2-L [53]	-	GTX 1080Ti	75.3	47.3
PP-Lite-T2 [54]	STDC1	GTX 1080Ti	74.9	143.6
STDC-Seg75 [17]	STDC1	GTX 1080Ti	75.3	**126.7**
CABiNet [55]	MobileNetV3	RTX 2080Ti	75.9	76.5
STDC-CT75	STDC1	RTX 2080Ti	**76.5**	122.6

**Table 7 sensors-23-06514-t007:** Comparisons with other mainstream methods on UAVid.

Model	Class IoU (%)								mIoU (%)
	Clutter	Building	Road	Tree	Low Veg.	Mov. car	Static Car	Human	
FCN-8s [34]	63.9	84.7	76.5	73.3	61.9	65.9	45.5	22.3	62.4
SegNet [56]	65.6	85.9	79.2	78.8	63.7	68.9	52.1	19.3	64.2
BiSeNet [16]	64.7	85.7	61.1	78.3	**77.3**	48.6	**63.4**	17.5	61.5
U-Net [35]	61.8	82.9	75.2	77.3	62.0	59.6	30.0	18.6	58.4
BiSeNetV2 [53]	61.2	81.6	77.1	76.0	61.3	66.4	38.5	15.4	59.7
DeepLabv3+ [49]	68.9	87.6	**82.2**	79.8	65.9	69.9	55.4	26.1	67.0
UNetFormer [57]	68.4	87.4	81.5	80.2	63.5	73.6	56.4	**31.0**	67.8
BANet [58]	66.6	85.4	80.7	78.9	62.1	69.3	52.8	21.0	64.6
STDC-Seg75 [17]	68.7	86.8	79.4	78.6	65.4	68.1	55.7	24.5	65.9
STDC-CT75	**69.2**	**88.5**	80.1	**80.4**	66.3	**73.8**	60.3	28.4	**68.4**

**Table 8 sensors-23-06514-t008:** Detailed specifications of the Jetson TX2 embedded system.

Items	Specification
CPU	Dual-Core NVIDIA Denver 2 64-Bit CPU Quad-Core ARM® Cortex®-A57 MPCore
GPU	256-core NVIDIA Pascal™ architecture GPU
Power	7.5 W/15 W
Memory	8GB 128-bit LPDDR4 Memory 1866 MHx − 59.7 GB/s
Storage	32 GB eMMC 5.1
Operating system(OS)	Linux for Tegra R28.1
AI Performance	1.33 TFLOPs

**Table 9 sensors-23-06514-t009:** Inference speed on TX2.

Model	mIoU (%)	Inference Time (ms)
U-Net	63.4	392.63
PSPNet	52.1	332.47
DeepLabv3+	57.4	253.78
STDC-Seg	65.1	**52.71**
STDC-CT	**67.3**	58.32

## Data Availability

The UAVid dataset can be found at https://uavid.nl/ (accessed on 22 May 2023). The Cityscapes dataset can be found at https://www.cityscapes-dataset.com/ (accessed on 22 May 2023). The UAV-City dataset in this study is available from the corresponding author upon request.

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
