# Peer review of "A Real-Time Semantic Segmentation Method Based on STDC-CT for Recognizing UAV Emergency Landing Zones"

_sensors, 2023, doi:10.3390/s23146514_

Round 1
Reviewer 1 Report
In the paper titled “A Real-time Semantic Segmentation Method Based on STDC-CT for Recognizing UAV Emergency Landing Zones” authors propose semantic segmentation method for UAVs. This paper discusses the importance of real-time recognition of emergency landing zones for unmanned aerial vehicles (UAVs) and proposes a semantic segmentation approach called the STDC-CT network. The STDC-CT network consists of three branches and focuses on addressing the challenges posed by complex backgrounds, diverse categories, and small targets in UAV aerial images. The experiments conducted on various datasets demonstrate that the STDC-CT method achieves a balance between segmentation accuracy and inference speed, improving the accuracy of small object segmentation while maintaining real-time performance on different hardware platforms.
Overall, the article is well written and the article needs minor organization/presentation changes with the following points.
· The authors provide a detailed overview of the proposed framework in the introduction. However, the organization of the article is missing.
· The contributions can be made precise. Currently, they have too much text.
· While the authors detail many related works in section 2, a table is necessary to compare them. Moreover, some discussion should be added to identify the research gap in existing literature.
· The labels in figure 4 are too small. Moreover, figure 6 is going out of borders.
· What is the trade-off between segmentation accuracy and complexity (time or space) of the proposed system?
· Some relevant references are missing such as,
Inspection of unmanned aerial vehicles in oil and gas industry: critical analysis of platforms, sensors, networking architecture, and path planning (Journal of Electronic Imaging)
N/A
Author Response
请参阅附件。

Reviewer 2 Report
In this paper, the authors designed a novel segmentation method for UAV image segmentation and created a new UAV landing dataset. The novel network structure consists of several modules and performs well on three datasets. Several ablation studies were conducted on the proposed method. Here are my comments to improve this paper before publication:
1) Some references could be added to Section 2.1 Conventional methods for recognizing UAV emergency landing zones to review traditional UAV landing zone detection methods.
2) The notations in Section 4.1 should be consistent with Figure 7. For example, the F_1 and F1 in Figure 7. Also, it is better to add important notations, such as T^t, into Figure 7 for a clearer explanation. Although the dimensions of variables are discussed in lines 322-326, it is better to explain the dimensions of variables such as F_i and T^t and the intermediate results.
3) Explanation of Figure 8 should be added.
4) In lines 346-347, the authors claim that the Gaussian Filter will blur high-frequency information, but they also state that it can separate the edges of small objects from noise. Edge information is also known as high-frequency information. As a result, the Gaussian filter tends to reduce the sharpness of boundaries rather than enhance them. Explanation or math proof should be provided.
5) In Section 4.4, lines 393-400, similar to Figure 7, the notations are suggested to mark in Figure 10, and explanations of notations should be discussed, for example, the notations used in the Eqs. (6), (7).
Overall, this paper is clear and easy to follow. The topic and novelty are interesting.
I have added my comments to the Comments and Suggestions for Authors Section.
